# Fragmentation Level Drives Local Fish Assemblage Diversity Patterns in Fragmented River Basins

Gustavo Díaz [1,2,*], Konrad Górski [3,4], Aliro Manosalva [1,2], Bárbara Toledo [1,2] and Evelyn Habit [1,2]

1. Departamento de Sistemas Acuáticos, Facultad de Ciencias Ambientales, Universidad de Concepción, Concepción 4070386, Chile
2. Centro de Ciencias Ambientales EULA-Chile, Universidad de Concepción, Concepción 4070386, Chile
3. Instituto de Ciencias Marinas y Limnológicas, Facultad de Ciencias, Universidad Austral de Chile, Valdivia 5090000, Chile
4. Facultad de Ciencias, Universidad Católica de la Santísima Concepción, Concepción 4030000, Chile
* Correspondence: gusdiaz@udec.cl

**Abstract:** Longitudinal connectivity is the main attribute of river ecosystems and is essential for the maintenance of aquatic biota. When longitudinal connectivity decreases in a river network, abundance of some fish species decreases, and local extinctions may occur. Such abundance decreases and extinctions affect local assemblage structure (alpha diversity) and result in a high degree of dissimilarity among local assemblages (higher beta diversity). Specific ecological mechanisms that are behind these biodiversity changes induced by connectivity loss remain poorly understood. Here, we assessed the effects of fragmentation at the local and basin level, as well as local environmental variables on local fish diversity patterns in eight Andean river basins in central Chile (32–39° S). The results indicated that fish assemblages inhabiting pool habitats in highly fragmented basins were characterized by significantly lower species richness and alpha diversity mainly driven by absence of fish species with high dispersion capacities. Our results highlight the importance of the effects of barrier cascades upstream as drivers of local native fish diversity. Sustainable hydropower development necessitates system scale planning of the placement of future barriers and should consider both local and basin scale biodiversity indicators.

**Keywords:** river connectivity; fish; Andean rivers; alpha diversity; Chile

## 1. Introduction

Anthropogenic use of aquatic resources has created stressors that have contributed to the progressive deterioration of freshwater ecosystems, compromising their conservation and ecological status [1,2]. Dudgeon et al. [3] proposed five major categories for freshwater ecosystem stressors (overexploitation, species introduction, habitat degradation, water contamination, and flow alteration). Often, flow alteration and habitat degradation could be related to establishment of barriers, because these disrupt natural gradients in river [4]. Indeed, hydropower plants function as barriers that alter the transport of water, sediments, organic matter, and nutrients, while also impeding movement patterns of freshwater biota [4,5]. Consequently, hydropower development is among the most significant drivers of physical and biological connectivity loss with detrimental consequences for the integrity and resilience of river ecosystems [6–8].

Biodiversity patterns of river ecosystems are strongly tied to the dendritic and hierarchical nature of river networks [9,10]. The structure of river networks composed of branches and nodes offers the biota restricted dispersal paths, so connectivity among aquatic habitats is essential, especially for strictly aquatic biota such as fish [11–13]. Indeed, barriers interrupt fish dispersal routes and modify natural environmental conditions of fish habitats, impeding completion of their life cycles [2,14]. Therefore, when longitudinal connectivity decreases in a river network, often, the abundance of fish species decreases

and local extinctions may occur (i.e., [15–17]). These effects in local fish assemblages may be driven directly by fragmentation, but also by changes in environmental conditions (e.g., water quality) and flow regime [18,19]. Differences among fish life history traits moderate responses to environmental changes that are reflected in local assemblage structure (alpha diversity) and also result in a high degree of dissimilarity among local assemblages (higher beta diversity) [20–22]. Loss of connectivity primarily affects migratory and wide-ranging species, as it alters species dispersion, the main assembly mechanism that determines spatial structuring and biodiversity patterns within the river network [17,20,23,24].

In Chile, Andean rivers are severely threatened by multiple anthropogenic stressors, including the exponential growth of hydropower development in recent decades [25–27]. Despite this, the number of hydropower plants is expected to increase significantly in the near future [27]. This hydropower boom has led to the loss of longitudinal connectivity with consequent modification of river networks and riverine habitats, which cause severe changes in fish genetic population connectivity, local abundances, and regional diversity patterns [20,28]. Nevertheless, specific ecological mechanisms that are behind these biodiversity changes remain poorly understood.

Here, we assessed the effects of fragmentation at the local and basin level, as well as local environmental variables on local native and non-native fish diversity patterns in eight river basins in a broad latitudinal gradient. The following hypotheses were evaluated: (i) The high river basin fragmentation level causes a decrease of local (alpha) diversity of fish assemblages; and (ii) dispersion capacity of each species moderates these local diversity responses to fragmentation.

## 2. Materials and Methods

### 2.1. Study Area

The study area comprises Chile's central zone and includes the Andean basins of the Aconcagua, Maipo, Rapel, Mataquito, Maule, Itata, Biobío, and Imperial rivers (32–39° S, Figure 1). They share characteristics such as a hydrological pluvio-nival regime, predominant climate [29,30], and are located in biodiversity hotspots [31]. The study area is also placed in the most densely populated and industrialized zone of Chile associated with high water demand for domestic and industrial use, as well as hydropower generation [25]. Consequently, river basins in this area are characterized by their high degree of fragmentation due to the large number of barriers, mainly hydropower plants [27].

### 2.2. Fieldwork

#### 2.2.1. Fish Sampling

Fieldwork was carried out between December 2015 and April 2016 (austral summer), with samples collected at 81 sites in eight river basins (Figure 1). The number of sites per basin varied depending on the number and location of barriers within the basin. If present, riffles (current speed $\geq$ 0.3 m/s) and pools (current speed < 0.3 m/s) were sampled in each basin at sites located upstream and downstream from each barrier. Overall, 79 riffles and 41 pools were sampled. Fish capture was carried out following standard methods used for bioassessment [32]. Each site, riffle, and pool habitat was sampled using a single-pass electrofishing approach with a Halltech HT-2000 (Guelph, ON, Canada) electric fishing equipment. In each habitat of each site we covered approximately 70 m in an upstream direction and executed between 20 and 30 min of active fishing. We applied similar sampling effort and obtained comparable fish assemblage samples from each habitat in each site. Collected fish were sedated with BZ-20 anesthetic, identified to species level using specialized fish identification keys [33–35], counted, and returned to their original habitats.

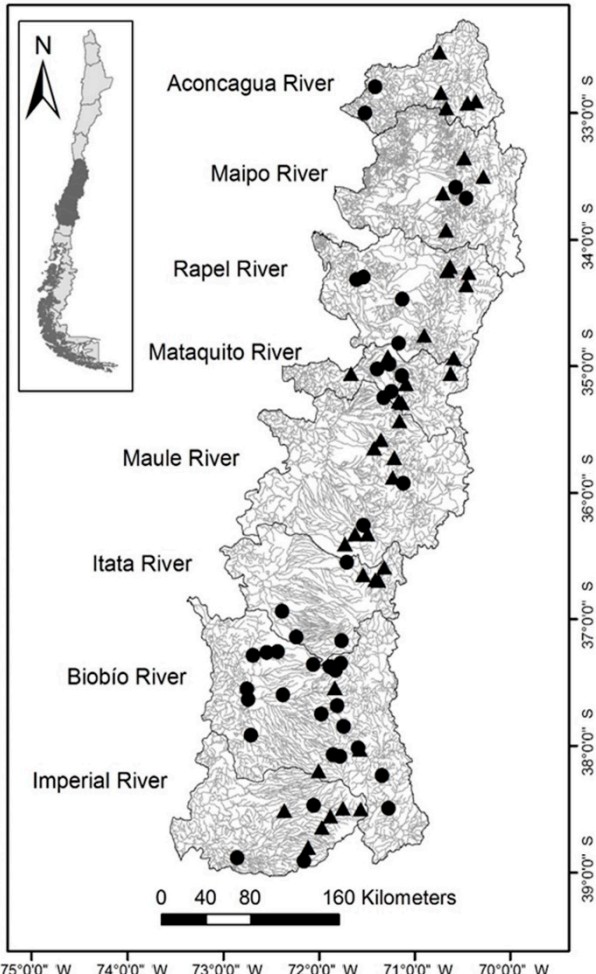

**Figure 1.** Location of sampling sites (triangles and dots) in the eight river basins that make up the study area. Dots correspond to sites with pool and riffles, whereas triangles correspond to sites with only riffles.

### 2.2.2. Local Environmental Variables

For the environmental characterization of riffles and pools, in situ temperature, pH, dissolved oxygen, conductivity, and total dissolved solids were measured using a multi-parametric probe Hanna HI-9828 (Woonsocket, RI, USA), while turbidity was measured using a turbidimeter Hanna HI-98703 (Woonsocket, RI, USA). Furthermore, in each habitat, current and depth velocity were measured using a Flow-Mate electromagnetic current meter Marsh McBirney Model 2000 (Loveland, CO, USA). Channel width at each site was estimated using satellite photographs generated in the summer season with Google Earth Pro Software (v.7.3).

### 2.3. Characterization of Fragmentation Status and Local Barriers

To characterize the level of fragmentation at the basin level, two fragmentation indices were used [24]. The longest fragment (LF) was calculated as the ratio between the longest fragment in the network and total length of the network within the basin, and fragmentation index (FI) considers the number and placement of barriers within the river network. LF reflects the longitude of the river network available for fish movement, and as such, this index only depends on the number of barriers, dismissing barrier characteristics, and its high values reflect low number of barriers in the river network [27]. Instead, FI represents the cumulative effect of barriers in the river connectivity network for fishes, considers the number of barriers, and their position in the river network, and its high values reflect high

fragmentation level. Both indices range between 0 and 1 [27]. Furthermore, we registered a set of variables that characterize the barrier immediately downstream of the sampling site to determine effects of the local fragmentation on fish assemblages for each sampling site. The following variables were used: type of barrier (natural waterfall > 20 m, irrigation canal-water diverting structure, run of river hydropower plant, reservoirs, or dams); location in the river network based on the Strahler order; elevation (m.a.s.l); operating time in years of existence of the barrier; and capacity, for hydropower plants measured as generation in MW. For natural waterfalls, a 200-year "operation time" was assumed. For barriers that do not correspond to hydropower plants, a capacity of 0.5 MW was assumed to reduce data heteroscedasticity. Finally, the number of barriers upstream and downstream of sampling site was determined.

*2.4. Statistical Analyses*

2.4.1. Local Diversity Analysis

Abundance matrices per sampling site were created for native and non-native fish in riffles and pools. Then, from these matrices, a set of local diversity measures was calculated for each fish assemblage, such as species richness (S), Shannon diversity (H'), and total abundance of individuals (N) using the "diversity" function of the *vegan* package [36]. Subsequently, the differences in local diversity (S, N, and H') were compared between the river basins, grouped based on their fragmentation level using Kruskal-Wallis non-parametric test.

2.4.2. Dispersion Capacity Analysis

Knowledge about the dispersion capacity of native Chilean fish species is still scarce, but body size and migratory behavior are recognized as decisive factors [30,34]. To analyze the dispersion capacity, a Principal Component Analysis (PCA) based on four indicators of fish dispersion (variables) representing known traits (ecological and biological) of each species was undertaken. This analysis, based on limited but known data, will be an approximation to identify species with high and low dispersion capacity and variables related to it. Variables used were "habitat use" (benthic or nektonic), "migratory behavior" (diadromous or resident species), "maximum body length recorded" for the species, and "range of environmental conditions", i.e., environmental variables that were registered at sampling sites where the species was found within this study. Prior to the PCA, variables were square-root transformed to reduce the effect of outliers. "Habitat use" and "migratory behavior" were considered binary characters, while "maximum body length recorded" and "range of environmental conditions" were continuous variables. The range of environmental conditions was estimated using the Outlying Mean Index (OMI) method [37] using the "niche" command of the *ade4* package [38]. To do this, we used the entire set of the environmental variables for the characterization of riffles and pools (temperature, pH, dissolved oxygen, conductivity and total dissolved solids, turbidity, channel width, current, and depth velocity).

2.4.3. Assemblage Structure Analyses

These analyses were conducted only for assemblages that presented significant differences in their local diversity measures between basins with high and low fragmentation levels. Thus, the analysis was based on presence-absence data when fragmentation effects were reflected in species richness, and abundances when fragmentation effects were reflected in total abundance or Shannon diversity. The contribution of each species to the assemblage was determined according to the fragmentation level ("high or low"). SIMPER analysis was performed using fish matrices prior to square-root transformation to decrease the heteroscedasticity. This analysis was performed using the "simper" command of the *vegan* package [36].

To examine how both local fragmentation and environmental variables explain the assemblage composition and structure of native fish, a distance-based redundancy analysis

(dbRDA; [39]) was executed. Models were performed using "forward" selection based on adjusted R² values and significance. Before performing this analysis, we examined the collinearity between variables, and highly related variables (R > 0.80) were removed. The estimation of total dissolved solids was deleted from further analyses due to high correlation with electrical conductivity. This analysis was run using the "capscale" and "ordiR2steep" function of the *vegan* package. All statistical analyses were executed with R software (v.3.2.2) [40].

## 3. Results

A total of 14 native species were found in riffle and pool habitats in the eight studied basins (Tables 1 and S1). *Diplomystes nahuelbutaensis* was the only native species found solely in riffle habitats, while *Odontesthes mauleanum* was the only native species present exclusively in pool habitats. A total of nine non-native species were captured. Among these, seven were captured in riffle habitats, while nine were in pool habitats. *Carassius auratus* and *Tinca tinca* were captured exclusively in pools.

**Table 1.** List of native and non-native fish species present in the study area, in addition to their habitat type, conservation category, and endemism. An "x" indicates presence in different habitat types.

|  | Species | Abbreviation | Endemic to Chile | Conservation Status | Riffle | Pool |
|---|---|---|---|---|---|---|
| Native | *Geotria australis* | Ga |  | Vulnerable | x | x |
|  | *Cheirodon pisciculus* | Cp | x | Vulnerable | x | x |
|  | *Cheirodon galusdae* | Cg | x | Vulnerable | x | x |
|  | *Nematogenys inermis* | Ni | x | Endangered | x | x |
|  | *Bullockia maldonadoi* | Bma | x | Endangered | x | x |
|  | *Trichomycterus areolatus* | Ta |  | Vulnerable | x | x |
|  | *Diplomystes nahuelbutaensis* | Dn | x | Endangered | x |  |
|  | *Diplomystes incognitus* | Di | x | Not assessed | x | x |
|  | *Galaxias maculatus* | Gm |  | Vulnerable/Least Concern * | x | x |
|  | *Basilichthys microlepidotus* | Bmi | x | Vulnerable/Near Threatened * | x | x |
|  | *Odontesthes mauleanum* | Oma | x | Vulnerable |  | x |
|  | *Percichthys trucha* | Pt |  | Near Threatened/ Least Concern ** | x | x |
|  | *Percilia gillissi* | Pg | x | Endangered | x | x |
|  | *Percilia irwini* | Pi | x | Endangered | x | x |
| Non-native | *Cyprinus carpio* | Cc |  |  | x | x |
|  | *Carassius auratus* | Ca |  |  |  | x |
|  | *Tinca tinca* | Tt |  |  |  | x |
|  | *Cheirodon interruptus* | Ci |  |  | x | x |
|  | *Salmo trutta* | St |  |  | x | x |
|  | *Oncorhynchus mykiss* | Omy |  |  | x | x |
|  | *Gambusia holbrooki* | Gh |  |  | x | x |
|  | *Cnesterodon decemmaculatus* | Cd |  |  | x | x |
|  | *Australoheros facetus* | Af |  |  | x | x |

* From Itata river basin to southern basins. ** Only for Itata, Biobío, and Imperial river basins.

Differences in fragmentation levels were recorded among studied basins based on fragmentation indices (Figure 2). LF showed a gradual increase from Rapel to Imperial river basin, whereas FI showed abrupt change with significantly higher values in the most fragmented basins. Based on these, Rapel, Biobío, Aconcagua, Maipo, and Maule river basins were classified as basins with a high degree of fragmentation, while Itata, Mataquito, and Imperial river basins with low level of fragmentation.

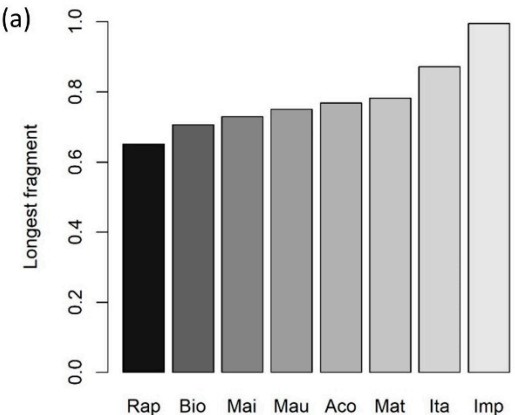

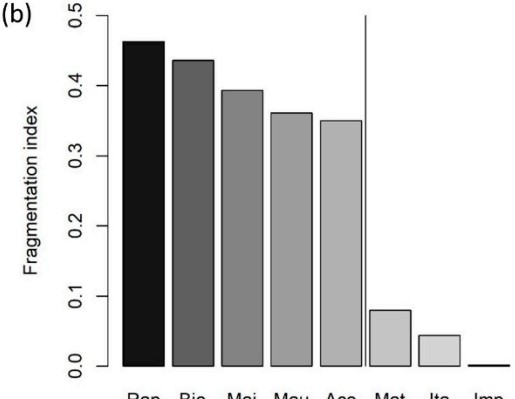

**Figure 2.** Fragmentation gradient based on (**a**) Longest fragment and (**b**) Fragmentation index values for eight studied basins. Vertical line shows the inflection point used to determine high and low fragmented basins categories. Aconcagua (Aco), Maipo (Mai), Rapel, (Rap), Mataquito (Mat), Maule (Mau), Itata (Ita), Biobío (Bio), and Imperial (Imp).

The diversity of native species showed significant differences between river basins with high and low levels of fragmentation (Figure 3). Specifically, species richness ($p = 0.021$) and Shannon diversity ($p = 0.008$) in pools, as well as total abundance ($p = 0.039$) in riffles, were significantly lower in highly fragmented river basins (Figure 3). In contrast, pools in highly fragmented basins exhibit higher values of species richness, total abundance, and Shannon diversity of non-native fish assemblages, however, these differences were not statistically significant (Figure 3).

Native fish species, *Percichthys trucha*, *Basilichthys microlepidotus*, *O. mauleanum*, *Galaxias maculatus,* and *Geotria australis,* were associated with nektonic habits, migratory behavior, and/or a broad range of environmental conditions, suggesting high dispersion capacities (Figure 4). Furthermore, *P. trucha*, *B. microlepidotus*, *O. mauleanum,* and *G. australis* were characterized by large body length (>12 cm), whereas *G. maculatus* by the broadest range of environmental conditions (Figure 4). *Percilia gillissi*, *P. irwini*, *Cheirodon galusdae*, and *C. pisciculus* that inhabited both pools and riffles were associated with resident behavior and small body size. Finally, both large (*N. inermis, D. incognitus, D. nahuelbutaensis*) and small (*Trichomycterus areolatus* and *Bullockia maldonadoi*) body-sized native catfish were associated with benthic habits (Figure 4).

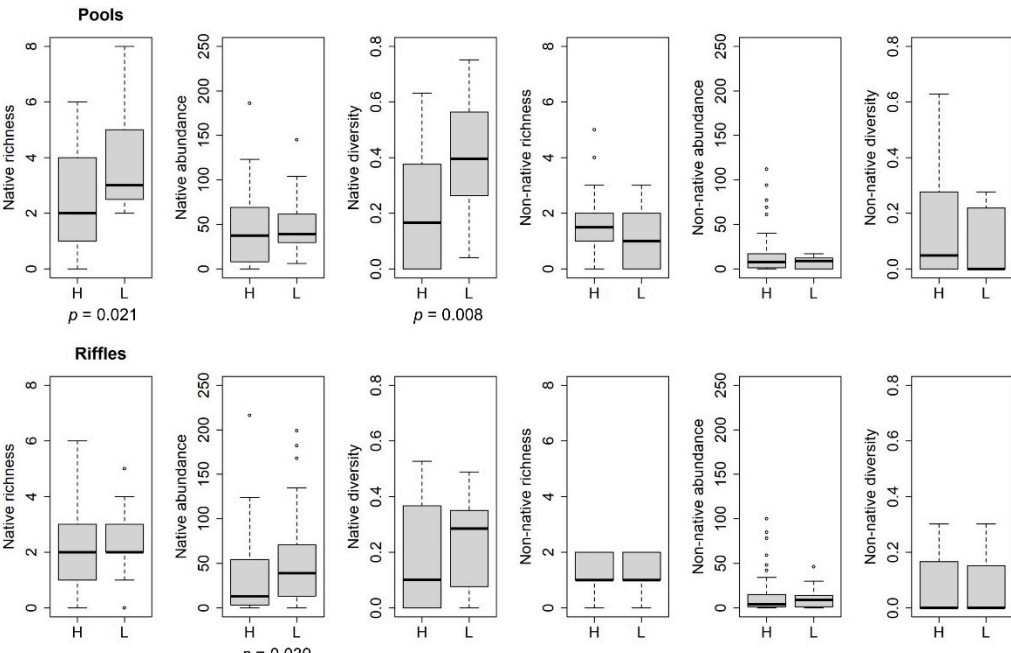

**Figure 3.** Diversity measures of native and non-native fish communities and the presence of statistically significant differences according to fragmentation level (L: Low; H: High) in riffle and pool habitats. Richness, S; diversity, H'; abundance, N.

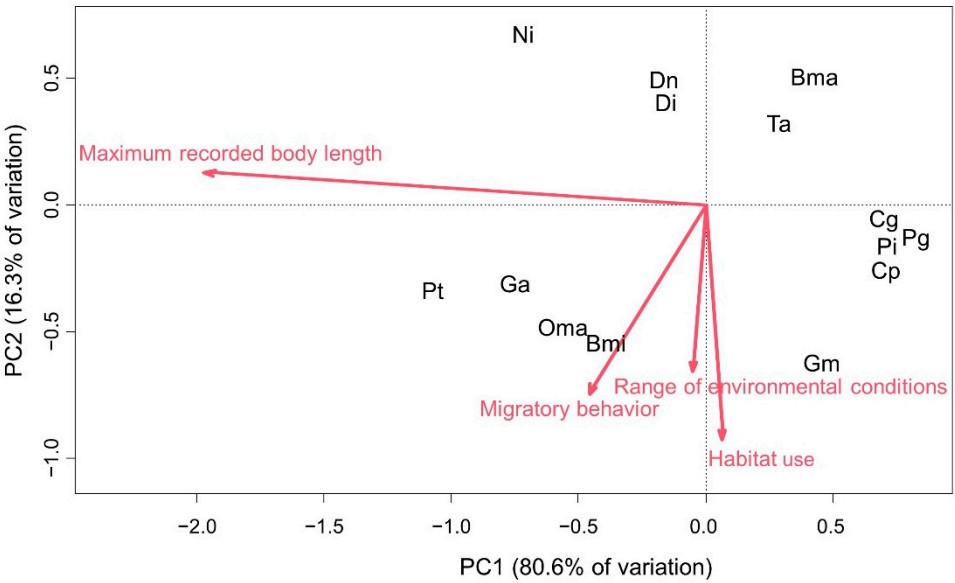

**Figure 4.** Principal component analysis based on biological and ecological traits that determine the dispersal capacity of the species present in the study area. Species near to the bottom on both sides of the ordination plot were considered to be characterized by high dispersal capacity, independent of the maximum recorded body length (left) or niche breadth (right). The figure shows the species (abbreviations in Table 1) and arrows that are related to each trait used in the analysis.

In pool habitats, migratory lamprey *G. australis* and resident large-bodied catfish *N. inermis* and *D. incognitus* were absent in river basins with a high level of fragmentation (Table 2). Furthermore, migratory *G. maculatus* and resident *P. gillissi* were significantly less abundant in these river basins (Table 3). In riffles, *G. australis* and *G. maculatus* were absent, whereas *P. gillissi* was significantly less abundant in river basins with high levels of fragmentation (Table 3).

**Table 2.** Frequency of occurrences of native species present in the eight basins studied according to level of fragmentation. High-dispersal capacity species appear in bold, while species with significant differences in occurrences are indicated with an asterisk (*).

| Habitat | Species | Frequency of Occurrence in Low-Fragmented Basins | Frequency of Occurrence in High-Fragmented Basins |
|---------|---------|-----|-----|
| Pools | *Percilia gillissi* * | 1.000 | 0.148 |
| | *Trichomycterus areolatus* | 0.545 | 0.629 |
| | *Percilia irwini* | 0.000 | 0.518 |
| | ***Percichthys trucha*** | 0.454 | 0.296 |
| | ***Basilichthys microlepidotus*** | 0.363 | 0.481 |
| | *Cheirodon pisciculus* | 0.272 | 0.111 |
| | ***Galaxias maculatus*** * | 0.363 | 0.111 |
| | *Cheirodon galusdae* | 0.181 | 0.259 |
| | ***Geotria australis*** * | 0.363 | 0.000 |
| | *Bullockia maldonadoi* * | 0.272 | 0.037 |
| | *Diplomystes incognitus* * | 0.090 | 0.000 |
| | ***Odontesthes mauleanum*** | 0.000 | 0.111 |
| | *Nematogenys inermis* * | 0.090 | 0.000 |

**Table 3.** Transformed average abundances of native species present in the eight basins studied according to level of fragmentation. Species were ordered according to absolute differences in abundances. High-dispersal capacity species appear in bold, while species with significant differences in average abundances are indicated with an asterisk (*).

| Habitat | Species | Average Abundance in Low-Fragmented Basins | Average Abundance in High-Fragmented Basins |
|---------|---------|-----|-----|
| Pools | *Percilia gillissi* * | 4.170 | 0.636 |
| | ***Basilichthys microlepidotus*** | 1.612 | 2.340 |
| | *Percilia irwini* | 0.000 | 2.190 |
| | *Trichomycterus areolatus* | 1.569 | 1.483 |
| | ***Percichthys trucha*** | 1.505 | 1.077 |
| | *Cheirodon pisciculus* | 0.765 | 0.379 |
| | *Cheirodon galusdae* | 0.639 | 0.829 |
| | ***Galaxias maculatus*** * | 1.002 | 0.168 |
| | ***Geotria australis*** * | 0.791 | 0.000 |
| | ***Odontesthes mauleanum*** | 0.000 | 0.490 |
| | *Bullockia maldonadoi* | 0.363 | 0.104 |
| | *Diplomystes incognitus* * | 0.090 | 0.000 |
| | *Nematogenys inermis* * | 0.128 | 0.000 |
| Riffles | *Trichomycterus areolatus* | 5.312 | 3.643 |
| | *Percilia gillissi* * | 3.247 | 1.009 |
| | *Percilia irwini* | 0.000 | 1.738 |
| | *Diplomystes incognitus* | 0.598 | 0.300 |
| | *Diplomystes nahuelbutaensis* | 0.000 | 0.514 |
| | ***Percichthys trucha*** | 0.473 | 0.241 |
| | *Bullockia maldonadoi* | 0.305 | 0.196 |
| | ***Basilichthys microlepidotus*** | 0.118 | 0.329 |
| | *Cheirodon galusdae* | 0.000 | 0.192 |
| | *Cheirodon pisciculus* | 0.104 | 0.065 |
| | *Nematogenys inermis* | 0.075 | 0.033 |
| | ***Geotria australis*** * | 0.086 | 0.000 |
| | ***Galaxias maculatus*** * | 0.150 | 0.000 |

Regression models based on fragmentation and environmental variables showed that elevation and number of barriers upstream explained the highest proportion of variation in species occurrence in pools (Table 4). Native species abundance in pools, in turn, were significantly influenced by elevation and conductivity, and in less extent by the

number of barriers upstream (Table 4). In riffles, a significant proportion of the variation in native species abundance was explained by elevation, number of barriers upstream, and conductivity (Table 4).

**Table 4.** Fragmentation and environmental variables at local level (site) that explain variation (model) in the native fish assemblage in eight river basins studied.

| Habitat | Approach | Model Variables | Cum $R^2$ Adj | Df | F | *p* |
|---------|----------|-----------------|---------------|-----|-----|-----|
| Pools | Abundance | Elevation | 0.051 | 1 | 3.027 | 0.002 |
| | | Conductivity | 0.093 | 1 | 2.668 | 0.002 |
| | | Number of barriers upstream | 0.136 | 1 | 2.706 | 0.002 |
| | | Type of barrier | 0.158 | 1 | 1.908 | 0.006 |
| | | All variables | 0.197 | | | |
| | Occurrence | Elevation | 0.040 | 1 | 2.556 | 0.002 |
| | | Number of barriers upstream | 0.068 | 1 | 2.080 | 0.004 |
| | | Conductivity | 0.098 | 1 | 2.162 | 0.002 |
| | | Capacity | 0.111 | 1 | 1.499 | 0.030 |
| | | All variables | 0.123 | | | |
| Riffles | Abundance | Elevation | 0.022 | 1 | 2.478 | 0.002 |
| | | Number of barriers upstream | 0.039 | 1 | 2.123 | 0.004 |
| | | Conductivity | 0.060 | 1 | 2.404 | 0.002 |
| | | Temperature | 0.067 | 1 | 1.431 | 0.048 |
| | | All variables | 0.095 | | | |

## 4. Discussion

The significant response of local native fish diversity to river fragmentation was documented in a broad latitudinal gradient. As such, native fish richness and diversity in pools was significantly lower in highly fragmented river basins, whereas in riffles, only native fish abundance was significantly lower in highly fragmented river basins. These responses in local habitats were mediated by the dispersion capacity of fish species composing fish assemblages.

Native species that were either migratory (*G. australis*) or resident large-bodied (*N. inermis* and *D. incognitus*) were absent in pools in highly fragmented basins. Indeed, migratory species were previously documented to be severely affected by fragmentation [41–43]. Furthermore, even though resident large-bodied catfish *D. incognitus* is also characterized by high dispersion capacities, genetic evidence for *D. incognitus* has shown that it can move through irrigation canals that interconnect different river basins [44]. In addition, morphologically and ecologically similar sister species *D. camposensis* has shown high within basin gene flow, broad home-ranges, and extensive movement in both upstream and downstream directions [45,46]. Knowledge about *N. inermis* biology and ecology is still limited, but results of the present study suggest that it may also be characterized by high dispersion capacity.

*Galaxias maculatus* and *P. gillissi* have shown significant declines in both frequency of occurrence and abundances in highly fragmented river basins. Migratory *G. maculatus* is characterized by facultative amphidromy, and it may also recruit in freshwater habitats [47,48]. As such, migratory populations of this species are probably most severely affected, hence the reduction of its frequency of occurrence. Furthermore, its abundance declines are probably associated with freshwater resident populations that have also been documented to experience extensive movements within freshwater habitats [49]. Resident small-bodied, *P. gillissi* has been described as a species with high genetic flow within non-fragmented river basins [46]. Furthermore, its hybridization with morphologically and ecologically similar sister species *P. irwini*, endemic to the highly fragmented Biobío River basin, mediated by population connection through irrigation canals, was recently documented [50]. Indeed, the sister species *P. irwini* was shown to experience movements of up to 30 km upstream within non-fragmented river reaches [51]. Moreover, it was shown

to experience significant abundance reduction and local extinctions of populations enclosed between two barriers [28].

Interestingly, we found no significant differences in local diversity and abundance of non-native species between river basins with high and low levels of fragmentation based on FI that takes into account position and number of barriers within the basin, despite the presence and high abundances of common non-native fish species of Andean river basins of the Chilean central zone (Table S2 and [20]). In contrast, beta diversity analyses for the same basins have shown significant increase of non-native species beta diversity in riffles with decreasing length of the longest non-fragmented section [20,52]. Furthermore, beta diversity responses seem to be driven by changes in abundance [17], whereas local (alpha) diversity changes of native species observed within the present study are driven by changes in both species richness and abundances. Similar responses were reported also in Gan River basin in China, where alpha diversity changes were driven by both loss of native species and also new records of non-native species [53].

Local (alpha) diversity changes seem to be significantly driven by elevation, conductivity, and the number of barriers upstream. As such, observed diversity changes are primarily driven by hydropower plants present in analyzed river basins, as these are the most frequent barriers [27]. Similar effects of the elevation and barrier density as the main driver of alpha diversity were shown for fish assemblages in Allier River in France [54]. In our study, observed effects are related to the number of dams located upstream of the sampling site that suggest cumulative dam effects on local habitats downstream [55]. Furthermore, these dam effects are reflected in local water quality changes, as shown by changes in conductivity. Indeed, most of the hydropower plants present in analyzed basins are run-of-river type, and these changes in water quality may therefore be explained by discharge reductions [25,56].

Our results suggest that hydropower plants are the primary drivers of changes in local native fish diversity in fragmented Andean river basins. These changes are driven by both route obstruction and changes in water quality and are mediated by dispersion capacities of different species. It was previously suggested that to preserve native fish assemblages in fragmented river systems, it is essential to maintain large free-flowing fragments within river networks [20]. Here, we highlight the importance of the effects of barrier cascades upstream as drivers of local native fish diversity. Indeed, multiple barriers and barrier cascades are expected to have severe cumulative effects on both the richness and abundance of native species [54]. Sustainable hydropower development necessitates system scale planning of the placement of future barriers and should consider both local and basin scale biodiversity indicators. Such planning should be based on proven solutions within hierarchical mitigation strategy: avoidance, minimization, restoration, and biodiversity offsets [5]. While some recent studies suggest planning strategies that may minimize some of the effects posed by barriers in Chilean Andean river basins [27,57], we postulate that the avoidance strategy should urgently be considered. This is valid especially in expected near future climate change scenarios, where up to 40% declines in river discharges in central Chile are expected [58], and as such, alternative energy sources should be evaluated.

**Supplementary Materials:** The following supporting information can be downloaded at: https: //www.mdpi.com/article/10.3390/d15030352/s1, Table S1: List of native and non-native fish species present in the study area and their conservation and endemism status. "x" denotes the presence of the species in the basins. Table S2: Transformed average abundances of non-native species present in the eight basins studied according to level of fragmentation. Species with significant differences in average abundances are indicated with an asterisk (*).

**Author Contributions:** Conceptualization, G.D., E.H. and K.G.; methodology, G.D. and K.G.; software, G.D.; validation, G.D., E.H. and K.G.; investigation, G.D., E.H., K.G., A.M. and B.T.; data curation, G.D., A.M. and B.T.; visualization, G.D., A.M. and B.T.; resources, E.H.; writing-original draft preparation, G.D., E.H. and K.G.; writing—review and editing, G.D., E.H., K.G., A.M. and B.T.; funding acquisition, E.H. All authors have read and agreed to the published version of the manuscript.

**Funding:** This research was funded by FONDECYT 1150154 and FONDECYT 1190647 to E.H. G.D. was funded by Beca Doctorado Nacional, ANID.

**Institutional Review Board Statement:** The capture of fishes for their identification and subsequent return to their environment were carried out under authorization of Subsecretaría de Pesca (SUB-PESCA/Resolución Exenta N°784), in addition to the bioethical committee of the University of Concepción (2015).

**Data Availability Statement:** Not applicable.

**Acknowledgments:** We thank Jorge González, Anaysa Elgueta, and Francisca Valenzuela for their help during the fieldwork. Jani Heino is acknowledged for valuable comments and suggestions on early draft of this manuscript.

**Conflicts of Interest:** The authors declare no conflict of interest.

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
