# Peer review of "Fragmentation Level Drives Local Fish Assemblage Diversity Patterns in Fragmented River Basins"

_diversity, doi:10.3390/d15030352_

Round 1
Reviewer 1 Report
This manuscript analyses the impact of dams and hydropower plants on native and invasive fish assemblages in Andean rivers. I believe that this is a useful and interesting paper. The authors analyze the data correctly from 81 sampling points, a big amount of samples.
There is only one relevant mistake, in my opinion. I need more explanation about how the authors carried out the electrofishing samplings. Besides, they use the number of collected fish as a variable, whereas this abundance should be relativized according to sampling effort, timing, and technical sampling.
capacities of different species.
Line 30: Well, not only Dudgeon et al., these are the five “official” principal threats that IUCN proposes, but never mind, do not worry about it.
Line 57: in Rodeles et al (2017), authors analyze how there is no information about fish assemblages in Spanish rivers up and down dams, as an example of this lack of understanding.
Line 81: please, explain the methodology better. How long were the surveys? Using gillnets to limit the sampling surface or not? How do you estimate the abundance?
Line 117: You cannot use absolute abundance but relative abundance because the number of collected fish depends on capture effort.
I believe that this paper would be useful for this research
Rodeles AA, Galicia D, Miranda R. 2017. Recommendations for monitoring freshwater fishes in river restoration plans: A wasted opportunity for assessing impact. Aquatic Conservation: Marine and Freshwater Ecosystems 27 (4), 880-885.
Author Response
We thank reviewer 1 for his comments and suggestions. Here we attached table of responses, with the changes requested to improve our article according to its scope and aim.

Reviewer 2 Report
The paper is interesting and was enjoyable to read. Understanding how fragmented rivers impact fish diversity is a relevant topic. However, this is not a particularly novel finding. Nevertheless, this paper could be of interest to readers.
One major concern I have is how some of the methods and analysis address the stated hypotheses. The authors hypothesis are only focused on how fragmentation impacts diversity. Yet, there is significant amount of effort spent on evaluating how environmental variables impact diversity. Environmental variables are also included in the first objective statement (line 58-560). Based on this it seems the environmental variables were assessed for no purpose. I suggest incorporating a hypothesis to be evaluated for environmental variables. Additionally, much of the analysis compares native and non-native species but this distinction is not mentioned in an objective, hypothesis, or introduction.
A second major concern is the application of the Principal Components Analysis. First, PCA is used to reduce the dimensionality of a data set by grouping correlated variables. This is often conducted when there are a large number of variables and a large number of variables that are correlated. The analysis used here seems to only use four variables. So I wonder if using PCA is overkill? Couldn't bivariate correlation of each species with environmental variables give the same answer and be simpler? Additionally, if PCA is used, there needs to be additional summary information about the results to adequately interpret as a reader. At minimum, Figure 4 should include points for each observation. PCA is very prone to a "horseshoe" effect. When this occurs, the PCA is not valid. In my experience, the "horseshoe" effect is most often found when using variables on different scales (e.g., dichotomous and continuous variables).
The introduction could be improved. As presented, it is only focused on fragmentation. But a significant about of effort is spent on analyzing the relationship between environmental variables and diversity. The introduction needs to be expanded to include environmental variables and their impact on diversity.
Line 38: "tied"?
Line 100-102: The analysis uses a fragmentation index which is described as including the number and placement of barriers within the river network. More details is needed. What is the range of values of this metric? How does the number and placement of barriers impact the index? A citation is provided but more information in the methods would be helpful.
Line 110: It seems the assignment of 0.5 MW was applied to natural barriers was chosen for statistical reasons only. This is not a good reason to make decisions. Larger hydropower plants will have a higher capacity. So a large natural barrier would be equivalent to a small hydropower plant?
Line 126: "range of environmental variables" is confusing. What range of variables are used in the analysis? Why is there a single variable for range of environmental variables but multiple other variables were measured?
Line 132: What habitat variables were used?
Line 148: Variables were removed if they were highly correlated. This is okay but how was one selected over the other?
Line 216-218: These conclusion are based on very low R-squared values. Yes, p-values were small but the effect size based on R-squared is very low. I would caution against making strong claims with cumulative R-squared < 0.20 and most variable specific R-squared < 0.10.
Line 243: needs a citation
Line 263-264. Citation 17 is a paper about how longest fragment size driving beta diversity (which is part of this study) but the statement with the reference is about changes in abundance driving beta diversity. This sentence needs a different reference.
Author Response
We thank reviewer 2 for his comments and suggestions. Here we attached table of responses, with the changes requested to improve our article according to its scope and aim.

Reviewer 3 Report
Dear authors,
I consider the article very well written with clear results and very well argued. For this reason, I would like to congratulate you and hope that the same article will be published very soon.
Author Response
We thank to reviewer 3 for your congratulations. We don´t have responses beacause he did not submit comments.
Round 2
Reviewer 2 Report
The authors have made significant revision to the manuscript and it has improved. However, I still have questions about interpreting the PCA analysis. The current figure 4 shows species and habitat/environmental variable scores. But they do not show the individual observations. The individual observations are needed to assess the horseshoe effect. Unless I am mistaken and the single species labels represent one observation, which I do not think they do. If the single species labels do not represent a single observation than a biplot with individual observations should be presented. If the species labels are single observations then the sample size seems small to draw inference. The general rule of thumb is 5-10 observations per variable. With four variables the minimum sample size would be 20 while 14 species labels are present. I understand adequate sample size can be an issue in ecological studies. And I believe the data and results are worth publishing. If there are only 14 observations then the small sample size must be addressed in the discussion to let the reader know of the limitations.
Author Response
Thank you for your comment. In the table of responses we answer your question and cover your concern.
